# Fabrication of Low Cost and Low Temperature Poly-Silicon Nanowire Sensor Arrays for Monolithic Three-Dimensional Integrated Circuits Applications

**DOI:** 10.3390/nano10122488

**Published:** 2020-12-11

**Authors:** Siqi Tang, Jiang Yan, Jing Zhang, Shuhua Wei, Qingzhu Zhang, Junjie Li, Min Fang, Shuang Zhang, Enyi Xiong, Yanrong Wang, Jianglan Yang, Zhaohao Zhang, Qianhui Wei, Huaxiang Yin, Wenwu Wang, Hailing Tu

**Affiliations:** 1School of Information Science and Technology, North China University of Technology, Beijing 100144, China; tangsiqi@ime.ac.cn (S.T.); zhangj@ncut.edu.cn (J.Z.); weishuhua@ncut.edu.cn (S.W.); fangmin@ime.ac.cn (M.F.); zhangshuang2020@ime.ac.cn (S.Z.); xiongenyi@ime.ac.cn (E.X.); wangyanrong@ncut.edu.cn (Y.W.); 2Advanced Integrated Circuits R&D Center, Institute of Microelectronic of the Chinese Academy of Sciences, Beijing 100029, China; zhangqingzhu@ime.ac.cn (Q.Z.); lijunjie@ime.ac.cn (J.L.); yinhuaxiang@ime.ac.cn (H.Y.); wangwenwu@ime.ac.cn (W.W.); 3State Key Laboratory of Advanced Materials for Smart Sensing, General Research Institute for Nonferrous Metals, Beijing 101402, China; yangjianglan955295@hotmail.com (J.Y.); tuhl@grinm.com (H.T.); 4School of Microelectronics, University of Chinese Academy of Sciences, Beijing 100049, China

**Keywords:** silicon nanowire (Si NW), monolithic three-dimensional integrated circuits (M3D-ICs), spacer image transfer (SIT), sensitivity

## Abstract

In this paper, the poly-Si nanowire (NW) field-effect transistor (FET) sensor arrays were fabricated by adopting low-temperature annealing (600 °C/30 s) and feasible spacer image transfer (SIT) processes for future monolithic three-dimensional integrated circuits (3D-ICs) applications. Compared with other fabrication methods of poly-Si NW sensors, the SIT process exhibits the characteristics of highly uniform poly-Si NW arrays with well-controlled morphology (about 25 nm in width and 35 nm in length). Conventional metal silicide and implantation techniques were introduced to reduce the parasitic resistance of source and drain (SD) and improve the conductivity. Therefore, the obtained sensors exhibit >10^6^ switching ratios and 965 mV/dec subthreshold swing (SS), which exhibits similar results compared with that of SOI Si NW sensors. However, the poly-Si NW FET sensors show the V_th_ shift as high as about 178 ± 1 mV/pH, which is five times larger than that of the SOI Si NW sensors. The fabricated poly-Si NW sensors with 600 °C/30 s processing temperature and good device performance provide feasibility for future monolithic three-dimensional integrated circuit (3D-IC) applications.

## 1. Introduction

In recent years, the application of semiconductor field-effect transistors (FET) sensors have attracted a lot of attention because of their ability to translate the interaction with target molecules on the FET surface to an electrical signal directly [1,2,3,4]. Silicon nanowires (Si NW) sensors have been considered as one of the most promising candidates for biochemical sensors [5,6,7,8,9], due to their large surface to volume (S/V) ratio, high sensitivity, and good biocompatibility [10,11]. In recent years, Si NW field-effect-transistor (FET) sensors have been used for the very high sensitivity detection of pH [12,13,14], gases [15,16,17] and DNA [18,19,20]. However, the conventional fabrication process with silicon-on-insulator (SOI) materials is complex and high-cost, and the nanometer patterns are usually formed by traditional low-efficiency electron beam lithography (EBL) process, which does not meet the demands of future mass production with a low cost and low efficiency. The spacer image transfer (SIT) process (also named self-aligned double or quadruple patterning) could achieve nanometer array with high efficiency and low cost, which are widely used in foundries [21,22,23]. In addition, there are no reports on the design or fabrication of Si NW sensors for monolithic three-dimensional integrated circuits (3D-ICs) application, which is one of the most convincing candidates for future application (see Figure 1). In order to achieve better performance of the system, the fabrication of top devices (Si NW sensor) usually need low-temperature processes to avoid a degradation on characteristics of bottom circuits (logic circuits and memory in Figure 1) [24]. However, there are no reports on the design or fabrication of Si NW sensor for monolithic 3D-ICs application.

In this paper, poly-silicon NW sensors with low cost and high efficiency are designed and fabricated using advanced spacer image transfer (SIT) [22,23,24] and low-temperature silicide techniques for monolithic 3D-IC application. The highest annealing temperature is not over 600 °C, which overcomes the problems of overheating the bottom transistor and the wires. The achieved poly-silicon NW sensors have good electrical properties, such as over six orders of magnitude in on-off ratio and 932 mV/dec of subthreshold swing (SS) by bias back-gate voltages.

## 2. Materials and Methods

Two types of Si NW sensors (poly-silicon and silicon-on-insulator (SOI)) were designed and fabricated, and the detailed fabrication flow is illustrated in Figure 2. The poly-silicon NW sensors were manufactured on p-type 200 mm Si (100) silicon wafers (see Figure 2a): Firstly, 145 nm SiO_2_ and 40 nm poly-silicon were deposited, respectively (see Figure 2b). The SOI Si NW sensors were manufactured on 200 mm SOI wafers featured with a 145-nm-thick buried oxide layer (BOX) and a 40-nm-thick top silicon layer. The fabrication process of the two types of devices was all the same in the flowing steps. During the fabrication of the Si NWs, a spacer image transfer (SIT) technology was chosen to form NW arrays patterns with high efficiency [25,26,27], and the detailed fabrication process flow is described as follows: sequential multi-layer SiO_2_/amorphous Si (α-Si)/Si_3_N_4_ films were deposited (see Figure 2c). Next, the conventional photolithography process (i-line) and dry etching processes were used to form rectangular arrays of Si_3_N_4_ and α-Si films (see Figure 2d). The top Si_3_N_4_ hard masks (HMs) were removed by a hot H_3_PO_4_ solution at 140 °C (see Figure 2e). A 30 nm Si_3_N_4_ film was deposited by plasma-enhanced chemical vapor deposition (PECVD) approach and then the corresponding Si_3_N_4_ reactive ion etching (RIE) was performed to form two SiN*_x_* spacers on both sides of α-Si (see Figure 2f,g). The α-Si material between two Si_3_N_4_ spacers was removed by tetramenthylammonium hydroxide (TMAH) (see Figure 2h). In order to obtain Si NW arrays, dry etching processes of SiO_2_ and Si were carried out, respectively (see Figure 2h). Afterward, the top HMs were removed using hot phosphoric acid and diluted hydrofluoroacid (DHF) solution, respectively (see Figure 2i). After the Si NW formation, a 5-nm-thick SiO_2_ was deposited on the Si NW followed by the deposition of a thick layer of Si_3_N_4_ (see Figure 2j,k). The Si_3_N_4_ film was etched by dry etching processes. A nickel platinum alloy (Ni_0.95_Pt_0.05_) was used to form metal silicide in the source and drain regions to reduce the parasitic of Si nanowires (see Figure 2l,m). Afterward, BF^2+^ ions with a heavy dose and low energy were implanted into the top silicided Si NWs and activated by low temperature rapid thermal annealing (RTA) to form Schottky barrier source and drain (SBSD) (about 600 °C/30 s annealing). For a better combination, the aluminum electrode was prepared by a sputtering process and the RIE process was performed (see Figure 2n,o). Subsequently, a layer of thick SiO_2_ was deposited, and the source drain contact holes were opened by photolithography and etching processes (see Figure 2p,q). Finally, the gate with different channel lengths (5, 10 and 15 μm) was defined by photolithography and the open gate trench of the sensor to expose the sensitive area was achieved by RIE processes. The bounding of SD contact was carried out and a layer of 20-nm-thick HfO_2_ was deposited on the surface of the device (see Figure 2r). Figure 2s is a schematic top view of the device. Except for the source drain and gate trench of the devices, other areas were covered by a thick SiO_2_, which helps to improve the stability and reliability of the solution.

The cross-sectional views and top views of the device’s structures were observed using S-5500 and S-4800 scanning electron microscopes (SEM, Hitachi, Tokyo, Japan), respectively. The cross-sectional profiles of the final device were performed using transmission electron microscopy (TEM, FEI Talos, Brno, Czech) and energy-dispersive X-ray spectroscopy (EDX, FEI Talos, Brno, Czech). The electrical characterization was performed using a B1500A (Keysight, Santa Rosa, CA, USA) semiconductor parameter analyzer.

## 3. Results and Discussion

The images of the fabricated poly-Si NWs sensors by the SIT process are shown in Figure 3. Figure 3a,b shows top views of poly-Si NW arrays by SEM measurement. As can been seen from the images, highly uniform poly-Si NW arrays without any landing pads are achieved. The achieved poly-Si NW arrays using the SIT approach have high efficiency, low cost and smaller sizes compared with that of fabricated using the EBL process. Figure 3c shows a cross-sectional view of poly-Si NWs. Contrasted with previous work [28,29], the dimensions and the morphology of the fabricated Si NW arrays are well controlled and extraordinarily small, theoretically providing higher sensitivity for the fabricated poly-Si NW sensor. Figure 3d,e shows top views of poly-Si NW sensors arrays. The length of the electrode is 2 mm; the gate lengths (L_Gs_) of poly-Si NW FETs are 5 μm, 10 μm, and 15 μm, respectively. The sensor current is increased and the device’s variations are reduced for the multi-channel poly-Si NW sensors.

Figure 4 shows the cross-sectional TEM image and the electron scattering spectrum (EDS) elemental mappings of the poly-Si NWs channel. According to the TEM image, the thicknesses of HfO_2_/SiO_2_ layers are 19.34 nm and 5.25 nm, respectively. The sizes of the well-controlled regular rectangle poly-Si NW is about 25 nm in width and 35 nm in length. Furthermore, the EDS analysis of Hf, O, poly-Si, and N elements demonstrates that the HfO_2_ and SiO_2_ films are very uniform and the interfaces are clear and flat without inter-diffusion. The well-controlled insulation layer could reduce the leakage current from liquid to device, providing a robust detection environment in the liquid.

Initial measurements of transfer and output curves (I_D_-V_G_ and I_D_-V_D_) were performed by applying a bias gate voltage. In the measurement of the I_D_-V_G_ curve, I_D_ was measured at constant drain voltages (V_D_ = −0.2 V, −1.2 V, −2.2 V), and the gate voltage was swept from 0 to −30 V. In the measurement of the I_D_-V_D_ curves, the drain current was measured at constant gate voltages (V_G_ from 0 to −20 V with a −2 V step), and the V_D_ was swept from 0 to −5 V with a −0.2 V step.

The I_D_-V_G_ and I_D_-V_D_ curves by bias gate voltages of the p-type poly-Si NW sensors are shown in Figure 5. Figure 5a–c shows typical I_D_-V_G_ curves of 5-μm-L_G_, 10-μm-L_G_ and 15-μm-L_G_ poly-Si NW devices, respectively. As can be seen from the images, smooth and uniform p-type MOSFET curves were achieved for the sensors fabricated at low temperature. The I_on_/I_off_ ratios of poly-Si NW devices with the 5-μm-L_G_, 10-μm-L_G_ and 15-μm-L_G_ are 5.68 × 10^6^, 2.84 × 10^6^ and 2.31 × 10^6^, respectively. The corresponding extracted values of subthreshold swing (SS) are estimated to be 1070 mV/dec, 965 mV/dec and 956 mV/dec, respectively. Figure 5d depicts the I_D_-V_D_ curves of poly-Si NW device 10-μm-L_G._ The drain current increases with increasing V_G_ bias, implying that the carrier’s concentration inside Si NWs can be linearly adjusted, and devices prepared at low temperatures exhibit good FET electrical performance.

The I_D_-V_G_ and I_D_-V_D_ curves of the SOI Si NW for 5-μm-L_G_, 10-μm-L_G_ and 15-μm-L_G_ are shown in Figure 6, respectively. As can be seen from the images, smooth p-type MOSFET curves are achieved for the sensors fabricate at low temperature. The I_on_/I_off_ ratios of 5-μm-L_G_, 10-μm-L_G_ and 15-μm-L_G_ SOI Si NW devices are 1.47 × 10^8^, 1.29 × 10^7^ and 6.34 × 10^4^, respectively, and extracted values of SSs are estimated to be 686 mV/dec, 767 mV/dec and 1120 mV/dec, respectively. Figure 6d depicts the I_D_-V_D_ curves of the 10-μm-L_G_ poly-Si NW device.

Figure 7 shows the extracted typical parameter comparison between the poly-Si NW and SOI Si NW sensors, e.g., the threshold voltage (V_th_), SS, on-stage current (I_on_) and I_on_/_off_ ratio, respectively. The V_th_s of 5-μm-L_G_, 10-μm-L_G_ and 15-μm-L_G_ poly-Si NW devices are −8.06 V, −8.125 V and −7.87 V, respectively. The V_th_ of 5-μm-L_G_, 10-μm-L_G_ and 15-μm-L_G_ SOI Si NW devices are −7.67 V, −7.95 V and −7.8 V, respectively. The values of V_th_s of poly-silicon devices with different L_G_s are similar to those of the SOI devices. The values of SSs of SOI devices are smaller than those of poly-Si devices and the I_on_/I_off_ is also larger. The performance of SOI devices is slightly better than that of low-temperature poly-silicon devices, which is caused by the monocrystalline silicon channel with a low channel resistance. The achieved results imply that the poly-Si NW sensors could be applied for future monolithic 3D-IC application.

Figure 8 shows the typical sensing characteristics of the poly-silicon nanowire sensors by analyzing different stranded pH solutions. In the measurement of the I_D_-V_G_ curve and the gate voltage was swept from 0 to −10 V by top solution (see Figure 8a inserted image). The detection principle is to convert the sensor surface potential change introduced by a different pH solution into the current change in the semiconductor Si NW channel. The actual amount of charges depends on the concentration of specific ions in the solution (the concentration of H^+^ ion in the manuscripts). Therefore, the pH of the solutions could modulate the surface charge of the insulator/semiconducting interface consequently, resulting in a shift of the threshold voltage. The scheme of the test using the top liquid gate is shown in the inserted figure of Figure 8a. Due to the change of film surface potential of the channel, the poly-silicon nanowire sensors exhibit V_th_ shifts (see Figure 8a). If the added solution is acidic (alkaline), the I_D_-V_G_ curve will shift to the right (left). After adding different pH buffers, the real-time response of I_D_ is shown in Figure 8b. Taking the buffer solution with pH = 7 as a reference, when the pH buffer is acidic, a positive charge is introduced and the current of the p-type poly-silicon nanowire sensor increases. When the pH buffer is alkaline, a negative charge is introduced and the current decreases. The results are consistent with the transfer curve of the p-type poly-silicon nanowire sensor increases. The extracted change values of V_th_ and I_D_ as a function of pH values are shown in Figure 8c,d, respectively. The changes of V_th_ and I_D_ have approximate linearity with the pH values, and the sensitivity as high as about 178 ± 1 mV/pH, which is caused by the small size in Si NW and large surface to volume ratio.

Figure 9 shows the typical sensing characteristics of the SOI nanowire sensors by analyzing different stranded pH solutions. In the measurement of the I_D_-V_G_ curve, the gate voltage was swept from 0 to −4 V by the top solution. Figure 9a shows that the threshold voltage shifts with the pH of the solution. The extracted threshold change is linear with the pH of the solution (see Figure 9b). A similar trend of V_th_ shift is obtained, but the values of changes are only about a fifth of the poly-silicon nanowire.

Table 1 shows system parameters comparison of the relevant reported results in recent years and our fabricated poly-silicon nanowire sensors. The SIT technique is used to form poly-silicon NW sensor arrays with 25 nm in width and 35 nm in length, which exhibits high efficiency and low cost than that of formed by EBL. Furthermore, the resistance and the device performance- the poly-silicon NW sensor is greatly improved by introducing SBSD techniques, which is attributed to achieve a larger I_on_/I_off_ ratio and smaller values of SSs. The results indicate that the poly-silicon NWs sensor fabricated by low-temperature annealing has much better characteristics than those of the sensors prepared by other methods, which attributed to its application for future monolithic 3D-IC applications.

## 4. Conclusions

In summary, low cost poly-Si NW sensors arrays are fabricated through an advanced SIT process with high efficiency than that formed by electron beam lithography, and the morphology of Si NW is well controlled with small sizes. Furthermore, a low-temperature flow (600 °C) with silicide and implantation is designed and carried out. Benefiting from the silicide and isolation processes, the poly-Si NW FET sensors show six orders of magnitude in switching ratio and a SS of 965 mV/dec, which is similar to the counterpart of the SOI Si NW sensor. In addition, the poly-Si NW FET sensors show the V_th_ shift as high as about 178 ± 1 mV/pH, which is five times larger than that of the SOI Si NW sensors. Therefore, the design and fabricated poly-Si NW sensor arrays approach provides a good option for its potential application of the monolithic 3D-ICs in the future.

## Figures and Tables

**Figure 1 nanomaterials-10-02488-f001:**
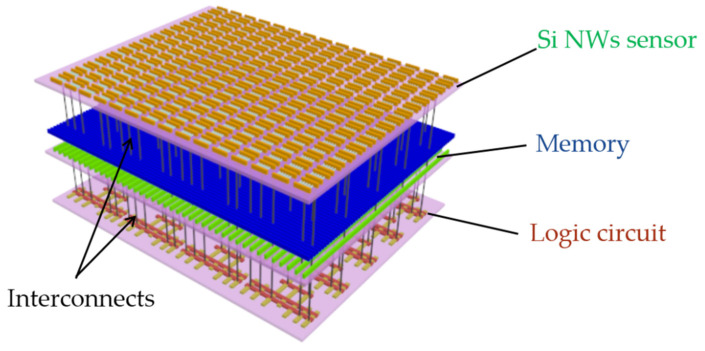
Illustration of monolithic three-dimensional integrated circuits (3D-ICs): bottom logic circuits; middle memory and top silicon nanowire (Si NW) sensors.

**Figure 2 nanomaterials-10-02488-f002:**
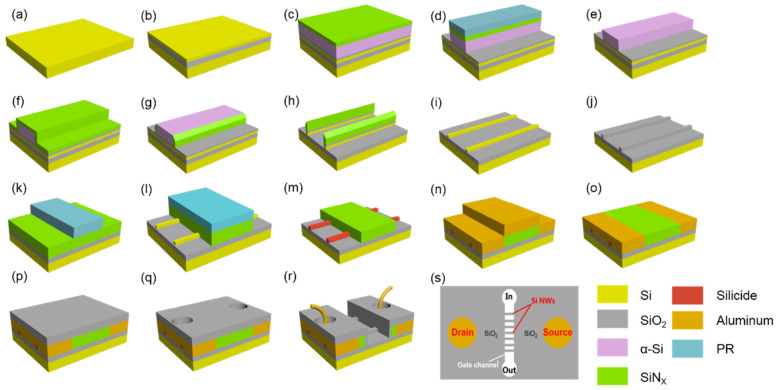
Fabrication flow of Si NW sensors. (**a**) 200 mm a p-type (100) silicon wafers, (**b**) deposition of SiO_2_ and α-Si films, (**c**) deposition of multi-layer SiO_2_/α-Si/Si_3_N_4_ films, (**d**) conventional lithography and reactive ion etching (RIE) of Si_3_N_4_/α-Si, (**e**) removal of photoresist (PR) and Si_3_N_4_ hard mask, (**f**) deposited of Si_3_N_4_ thin film, (**g**) anisotropic RIE of Si_3_N_4_ to form nanometer Si_3_N_4_ spacers, (**h**) remove the α-Si and RIE SiO_2_ and Si films, (**i**) removal of top hard masks (HMs), (**j**) deposition of 5 nm SiO_2_ film, (**k**) deposition of Si_3_N_4_ and i-line lithography, (**l**) RIE of Si_3_N_4_, (**m**) RIE of Si_3_N_4_ and forms metal silicide, (**n**) sputter metal, (**o**) RIE of metal, (**p**) deposition of HfO_2_ film, (**q**) formation of source and drain contact hole, (**r**) formation of Si NW channel and bonding and (**s**) top view of the designed Si NW sensors.

**Figure 3 nanomaterials-10-02488-f003:**
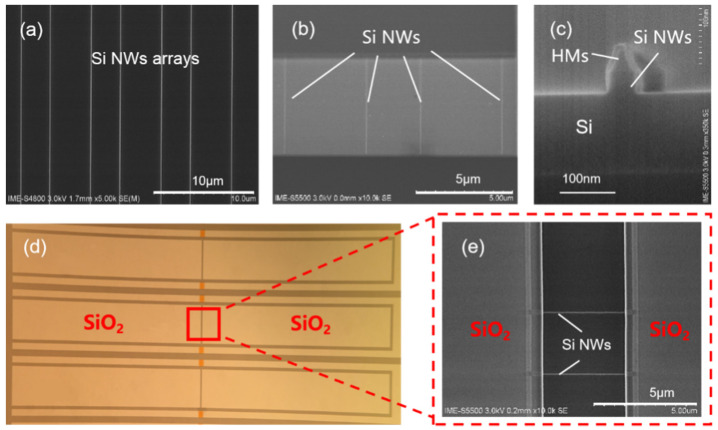
Images of poly-Si NW sensors. (**a**,**b**) SEM images of poly-Si NW array in top view, (**c**) cross-sectional SEM image of poly-Si NWs, (**d**) top view of poly-Si NW arrays sensor by optical microscope, (**e**) SEM image of the final poly-Si NW arrays sensor in top view.

**Figure 4 nanomaterials-10-02488-f004:**
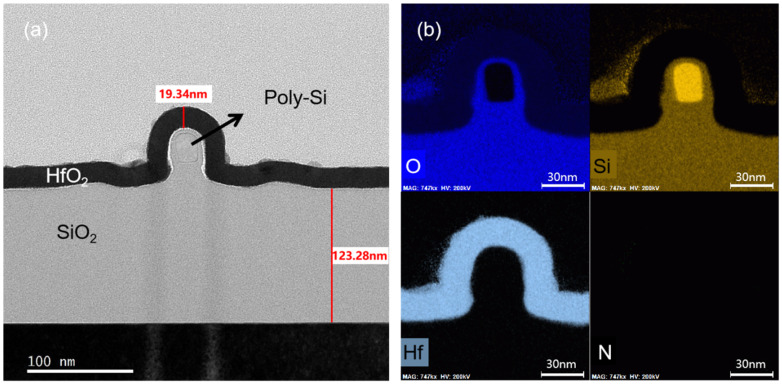
(**a**) TEM image of poly-Si NW channel of the device, (**b**) EDS elemental mappings of O, Si, Hf and N, respectively.

**Figure 5 nanomaterials-10-02488-f005:**
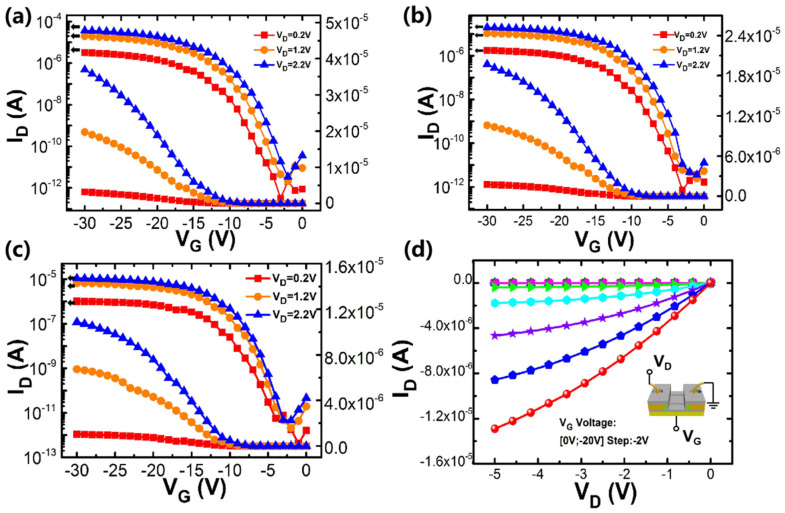
(**a**–**c**) I_D_-V_G_ curves of the fabricated 5-μm-L_G_, 10-μm-L_G_ and 15-μm-L_G_ p-type poly-Si NW sensors by back bias gate voltage, (**d**) typical I_D_-V_D_ of the fabricated 10-μm-L_G_ p-type poly-Si NW sensors.

**Figure 6 nanomaterials-10-02488-f006:**
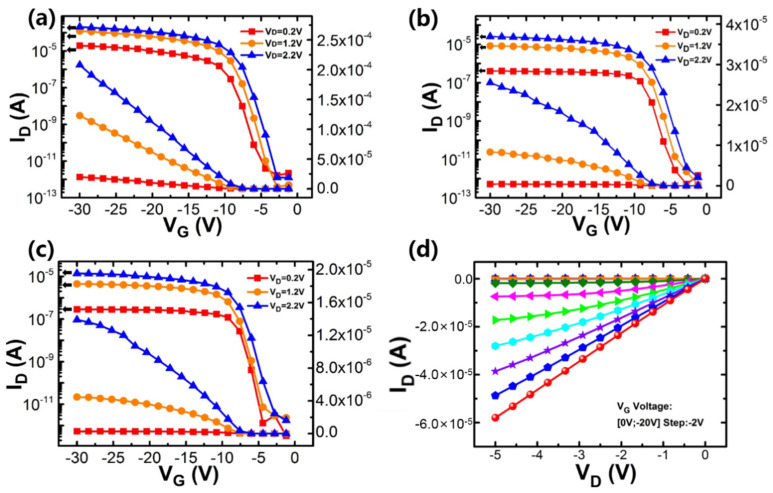
(**a**–**c**) typical I_D_-V_G_ curves of the fabricated 5-μm-L_G_, 10-μm-L_G_ and 15-μm-L_G_ p-type SOI Si NW sensors by back gate bias, respectively, (**d**) typical I_D_-V_D_ curves of the fabricated 10-μm-L_G_ p-type SOI Si NW sensors.

**Figure 7 nanomaterials-10-02488-f007:**
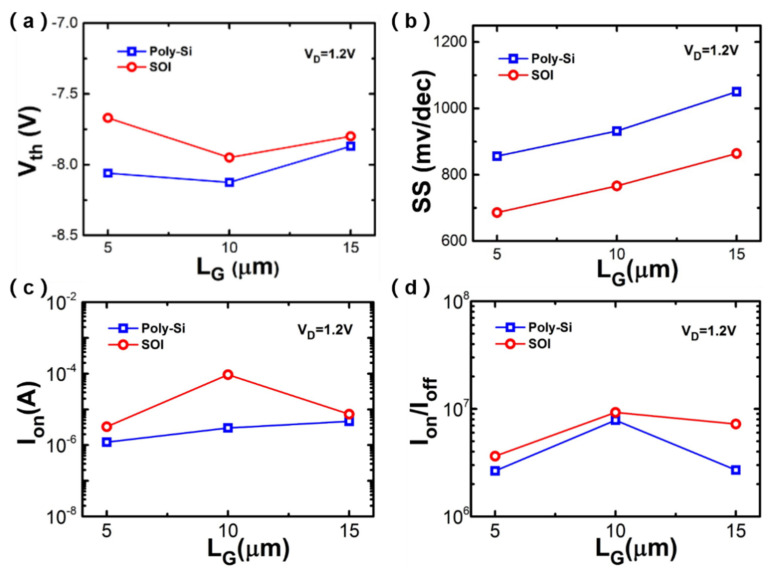
Comparison between low-temperature poly-silicon devices and SOI Si NW devices. (**a**) SS, (**b**) V_th_, (**c**) I_on_ and (**d**) I_on_/I_off_ ratio.

**Figure 8 nanomaterials-10-02488-f008:**
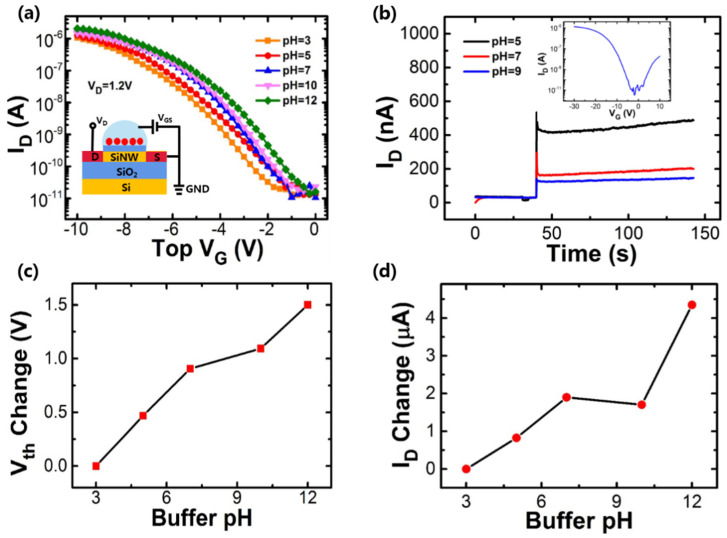
(**a**) The transfer curves of the poly-Si NW sensor by analyzing different pH solutions, (**b**) real-time response of I_D_ when adding different pH solutions, the inserted figure the transfer curves of the bias back gate voltage, (**c**,**d**) threshold voltage and drain current change under different pH solutions.

**Figure 9 nanomaterials-10-02488-f009:**
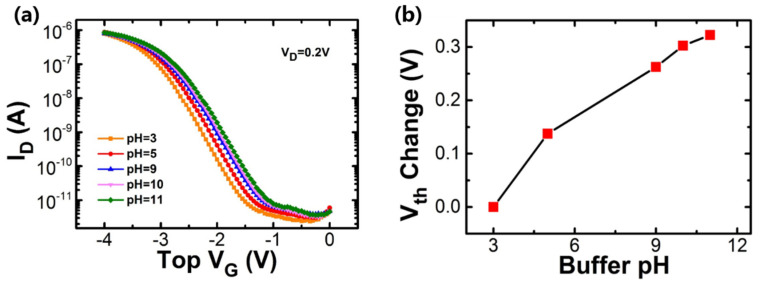
(**a**) The transfer curves of the silicon-on-insulator (SOI) sensor with different pH solutions, (**b**) threshold voltage under different pH solutions.

**Table 1 nanomaterials-10-02488-t001:** A comparison of silicon nanowire sensors in recent years.

	2011 [30]	2012 [31]	2014 [32]	2016 [33]	2020 [34]	This Work	
Insulation material	SiO_2_	SiN*_x_*	SiO_2_	SiO_2_/Si_3_N_4_	SiO_2_/Si_3_N_4_/SiO_2_	SiO_2_	SOI
Insulation thickness	100 nm	-	80 nm	50/65 nm	-/150/7 nm	145 nm	
NWs material	Si	Si	poly-Si	poly-Si	poly-Si	poly-Si	Si
Si NWs fabrication approach	VLS	RIE	sidewall spacer	RIE	EBL	SIT	SIT
Processing temperature	-	-	-	600 °C	1050 °C	600 °C	600 °C
NWs size	~90 nm	-	-	~40 nm	40~50 nm	~30 nm	~30 nm
L_G_	-	-	100 nm	10 μm	400 nm	10 μm	10 μm
I_on_/I_off_	~10^5^	~10^4^	-	2.03 × 10^5^	2.5 × 10^5^	2.84 × 10^6^	1.29 × 10^7^
SS (mV/dec)	2500	2300–3000	-	975	1030	965	767
V_th_ change (V)	-	-	0.087	0.0437	-	0.178	0.0688

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
