# Peer review of "Fabrication of Low Cost and Low Temperature Poly-Silicon Nanowire Sensor Arrays for Monolithic Three-Dimensional Integrated Circuits Applications"

_nanomaterials, 2020, doi:10.3390/nano10122488_

Round 1

Reviewer 1 Report

Nice scientific work poorly presented:

  1. Abstract should include quantified results, e.g. values for obtained subthreshold swing given in result chapter.
  2. In Chapter 2. Materials and Methods and in Figure 2 it is incomprihensive to understand how the poly-Si and the SOI variants are made differently. Figure 2 (s) too small with almost unreadable small text for and does not identify source and drain.
  3. In chapter 3. Results, the HfO2 layer is unexpectedly described - description should be moved to Chapter 2.
  4. The result chapter lacks ph immersion results with SOI version that should be included, and result comparison between poly-Si and SOI versions when immersed in ph solutions should be included.
  5. In Table 5: Where is the SOI version? Either it is not there or not properly identified.
  6. Discussions are included in the Result chapter. It is recommended to only list the objective results and put the more subjective discussions in a separate chapter "Results"
  7. In discussions of the results, avoid "gambling words" with no quantification and therefore no scientific value, like "indicate", "little" and "may be"
  8. In conclusion chapter: Where is the SOI version? Avoid the "gambling word"  "good" that has no quantification and therefore without scientific value.

With a major makeover this can be a good paper.

Reviewer 2 Report

The authors report on the fabrication of poly-silicon nanowire arrays and their use as transistors and sensors. The presented work appears to be technically solid and the described fabrication process is interesting for the community. Several points should be addressed before publication:

  • Line 26, rephrase: Due to the adopted the optimized process, …
  • Line 38-40, meaning unclear: Although nanowire … , achieved very high sensitivities.
  • The authors should comment on the thermal budget in CMOS processing. The use of 600degC might be problematic and limit the compatibility of the described process.
  • The authors should comment on the kinetics of the drain current increase after a change in pH level. How is the initial steep increase and subsequent decrease explained?
  • The authors are advised to provide data points for more different pH levels so that the reader can assess the dependence of Vth and Id on pH.
  • The authors are advised to compare their sensing performance with nanowire-based pH sensors reported in literature.

Author Response

Thank you very much for your comments. I have uploaded the review comments in the attachment. Please see the attachment.

Reviewer 3 Report

I am delighted to provide a review report on the manuscript of Tang et al titled “Investigation of Poly-silicon Nanowire Sensor Arrays with Low Temperature, Low Cost and High Efficiency for Future Monolithic 3D-IC Application” submitted for possible publication in nanomaterials (the journal). In their manuscript, they attempted to fabricate high efficiency poly-silicon NW sensors with facile yet low cost techniques by particularly using low-temperature advanced spacer image transfer (SIT). Their poly-silicon NW sensors showed good electrical properties, such as over 6 orders of magnitude in on-off ratio and 932 mv/dec of subthreshold swing (SS) by bias back-gate voltage. Overall, the device structure and material morphology demonstrated an option to be adopted in three-dimensional integrated circuits (3D-IC) applications. To the best of my knowledge, the materials employed and the technique of SIT are not new nor novel, but the concept to utilise Poly-silicon Nanowire Sensor Arrays in Monolithic 3D-IC Application is interesting. I would recommend to publish this article in the event the authors can make it better by looking through my minor comments below.

I found the title lengthy and having many buzz words like “High efficiency”, “future”. I would strongly recommend a short easily abstracted title with key terms dressing the article. In that sense, the word “investigation” can be dropped, simply because any research article attempts to investigate any way. Also avoid abbreviations like “3D-IC”

I recommend the authors to make the abstract quantitatively exact.. for example when they say “……with a smaller size” they should add the actual size in brackets say “….with a smaller size (xx units)”  What are the low processing temperatures? Can the  good device performance figures of merit be given here also? In the whole abstract only one figure of performance was given of “..exhibit good switching ratios (>106)”. Therefore, please revisit the abstract and amend accordingly.

Can the authors also detail or provide some literature about other fabrication techniques beyond advanced spacer image transfer (SIT) and why they specifically targeted SIT?  There seems to sit this gap between paragraph 1 and paragraph 2 of the introduction section

 Also, grammar and spellings should be polished. For example "souse and drain ?" in the abstract

Author Response

(The authors gave the same response as above.)

Round 2

Reviewer 1 Report

Your revised manuscripts is good, having followed up most of the comments given in review 1. It is regretted that you did not follow up:"6. Discussions are included in the Result chapter. It is recommended to only list the objective results and put the more subjective discussions in a separate chapter "Results"